# Surface Modification Enabling Reproducible Cantilever Functionalization for Industrial Gas Sensors

**DOI:** 10.3390/s21186041

**Published:** 2021-09-09

**Authors:** Daniel Mamou, Lawrence Nsubuga, Tatiana Lisboa Marcondes, Simon Overgaard Høegh, Jeanette Hvam, Florian Niekiel, Fabian Lofink, Horst-Günter Rubahn, Roana de Oliveira Hansen

**Affiliations:** 1NanoSYD Center, Mads Clausen Institute, University of Southern Denmark, 6400 Sønderborg, Denmark; jimam19@student.sdu.dk (D.M.); lawrence@mci.sdu.dk (L.N.); tlisboa@mci.sdu.dk (T.L.M.); rubahn@mci.sdu.dk (H.-G.R.); 2AmiNIC ApS, 5500 Middelfart, Denmark; simonh@mci.sdu.dk (S.O.H.); jh@aminic.dk (J.H.); 3Fraunhofer Institute for Silicon Technology, 25524 Itzehoe, Germany; florian.niekiel@isit.fraunhofer.de (F.N.); fabian.lofink@isit.fraunhofer.de (F.L.)

**Keywords:** cantilever gas sensing, cadaverine sensing, surface energy, surface modification

## Abstract

Micro-cantilever sensors are a known reliable tool for gas sensing in industrial applications. We have demonstrated the application of cantilever sensors on the detection of a meat freshness volatile biomarker (cadaverine), for determination of meat and fish precise expiration dates. For achieving correct target selectivity, the cantilevers need to be functionalized with a cadaverine-selective binder, based on a cyclam-derivative. Cantilever surface properties such as surface energy strongly influence the binder morphology and material clustering and, therefore, target binding. In this paper, we explore how chemical and physical surface treatments influence cantilever surface, binder morphology/clustering and binding capabilities. Sensor measurements with non-controlled surface properties are presented, followed by investigations on the binder morphology versus surface energy and cadaverine capture. We demonstrated a method for hindering binder crystallization on functionalized surfaces, leading to reproducible target capture. The results show that cantilever surface treatment is a promising method for achieving a high degree of functionalization reproducibility for industrial cantilever sensors, by controlling binder morphology and uniformity.

## 1. Introduction

Micro-scaled cantilever sensors with functionalized layers for gas sensors are a flexible and well-established technology [1]. When a gas molecule binds to a microcantilever, it increases the mass of the cantilever beam and thereby reduces its mechanical resonance frequencies. Therefore, the presence of few gas molecules is reflected by changes in the resonance frequencies being measured as impedance output through actuation of the piezoelectric material [2,3]. Due to the flexibility of adapting the functionalized layer to the target gas, there are unending possibilities for application of these sensors into a wide range of fields. We have previously studied its application for cadaverine (meat freshness biomarker) sensing and demonstrated the application of micro-cantilever sensors for this use [4,5,6,7,8].

Today, households, restaurants, catering and food stores rely on the printed expiration date, which is based on general prediction curves for meat degradation. For example, for pork cuts at 5 °C under aerobic conditions, the prediction curve says 8 days (+/−) 3 days. Therefore, to be on the safe side, expiration dates are set to 5 days. The only means to control actual freshness of meat products are microbiological tests conducted at external laboratories, requiring shipping of samples, growing and counting bacteria. Such tests are seldomly carried out, since they are expensive (app. 335 €/sample), time consuming (48 h) and do not predict the exact expiration date. There are other analysis methods available [9,10], but they all need to be carried out at external laboratories, e.g., measurement of cadaverine levels using gas chromatography mass spectrometry [11]. Cadaverine is a volatile biogenic amine providing an exact measure for meat spoilage level. The amount of cadaverine gas increases in a predictable way over time depending on the type of meat [12,13,14,15,16,17,18,19].

For achieving correct target selectivity, the cantilevers need to be functionalized with a cadaverine-selective binder, based on a cyclam-derivative. As reported on our previous publications regarding surface functionalization with cyclam [6,7,8], cyclam crystallization has been a characteristic with several different functionalization parameters and conditions. Cadaverine absorption by cyclam clusters has not been an issue in the proof-of-concept and laboratory tests since absorption is present and results in a clear change of resonance frequency. However, when we think of upscaling the cantilevers into the industrial scale, cyclam morphology on functionalized surfaces comes into play with a crucial role, since crystal structures are forming randomly, rendering functionalization difficult to reproduce in a large number of cantilevers. Investigations on the influence of functionalization solvents on the crystal formation indicates an important influence on the solvent polarity index for the resulting crystal sizes [6], indicating that high cyclam solubility in a solution allows for a dispersed material surface distribution upon functionalization, requiring higher diffusion energy for cyclam clustering into large crystals, via a nucleation-growth mechanism. Wang, Y. et al. presents the crystallization mechanisms, including high resolution imaging of the cyclam crystals.

Morphological changes on the binder and cyclam clustering/crystallization can occur in different cantilevers, which lead to distinct result outputs for distinct sensors. Industrial applications require a large number of identical sensors with similar binder morphologies and consequently, a reliable sensor output. Therefore, a need for cantilever-binder morphology standardization is required and important when upscaling. As described above, a proper material distribution on the surface hinders crystal formation. That has been shown to be minimized via proper cyclam solubility. Surface functionalization morphology and uniformity strongly depends on surface properties such as surface energy and surface tension, and methods for chemical and physical surface treatments have been applied to tune functionalization [20,21,22,23]. One of the main issues encountered with cyclam-derivative functionalization was cyclam clustering in the binder drop, which leads to an uncontrollable non-uniform layer and distinct analyte capture for different samples [7,8]. For achieving reliable and reproducible results in industrial applications, cyclam clustering needs to be avoided.

Our hypothesis is that by modifying the properties of the cantilever surfaces, an improved distribution of the fully dissolved cyclam solution across the surface can be achieved, reducing material clustering significantly.

In this study, we investigate both physical and chemical treatments on SiO_2_ and Si_3_N_4_ surfaces, the changes on surface energies and how the cadaverine selective cyclam-derivative binder morphology is influenced by the surface treatment and surface properties. Our hypothesis is that cyclam clustering, resulting in a non-uniform functionalization layer, arises from high internal stress in a functionalization film formed from drops with a high contact angle (on a hydrophobic surface), and therefore, a hydrophilic surface might result in a thin film with lower internal stress by ensuring a higher contact area between the drop and surface, resulting in a more uniform functionalization layer and enabling industrial reliability.

## 2. Materials and Methods

Silicon microcantilevers coated with Si_3_N_4_ (17 µm thick, 1 mm long and 1 mm wide) were fabricated by defining the cantilever area on a silicon wafer by photolithography, and subsequent reactive plasma induced ion etching, at the cleanroom facilities of Fraunhofer Institute for Silicon Technologies. A layer of piezoelectric Aluminum Nitride (AlN) is sputtered on the cantilevers and contacted by electrodes deposited via electron beam evaporation. The cantilever is wire-bonded on a printed circuit board and interfaced with an impedance analyzer electronic circuit on a hand-held device designed for cadaverine detection and meat freshness evaluation, as described in [4].

A 0.15 µL solution containing a cadaverine selective cyclam-based binder is applied on two cantilever surfaces via a high-precision syringe using the Model OCA 20—Automatic Contact Angle Measuring and Contour Analysis System from DataPhysics. Cadaverine measurements from the cantilevers exposed to salmon were extracted and compared.

Silicon wafers coated with 200 nm SiO_2_ or 200 nm Si_3_N_4_ were cut into 1 × 1 mm pieces with a dicing saw. The samples were cleaned with Acetone, Isopropanol, Deionized water and dried with Nitrogen blow. One of the following chemical and physical treatments were applied to 5 samples (SiO_2_ or Si_3_N_4_):

Plasma treatment: Samples were submitted to oxygen plasma for 10, 20 and 30 s with a power of 10.2 W, which is preset on the available oxygen plasma chamber. No significant changes on the surface properties were observed.

Annealing: Samples were annealed at 100 °C, 200 °C, 300 °C or 400 °C on a hotplate at ambient conditions.

PBS-tween treatment: PBS-tween solution was drop-casted on the surface and dried out on a hotplate at 100 °C for 10 min.

Surface nanoparticle dispersion: A solution of silver nanoparticles (100 nm diameter) suspended in a tetraoctylammonium bromide-based solution was drop-casted on the surface and dried out on a hotplate at 100 °C for 10 min.

Following this, the surface modified samples were characterized by extracting the surface energies based on three different liquid contact angles as references: water, ethylene glycol and methyleneiodide, using a contact angle setup from DataPhysics. The surface energies were extracted by using the software from DataPhysics.

Cadaverine binder solutions were prepared with a cyclam derivative, as detailed in reference [6]. The solutions were applied on the surface treated samples, and the contact angle of the binder solution interacting with the surface was extracted. The samples were analyzed to find the most reproducible interaction between binder and surface.

Finally, standardized binder was exposed to synthetic cadaverine for 2 min, and its morphology was studied to determine the degree of morphological change. The binder morphology before and after cadaverine exposure was extracted by standard optical microscopy.

## 3. Results and Discussion

### 3.1. Sensor Readings without Binder Morphology Standardization

Micro-cantilevers with a piezoelectric layer (Aluminum Nitride) were functionalized with a cadaverine selective binder and integrated into the hand-held electronic nose. The electronic nose head comes in tight contact with a meat sample and sniffs the cadaverine gas into the device. The hand-held device contains the appropriate electronics and embedded systems to actuate the cantilever and sense the resonance frequency shifts due to cadaverine absorption by the cantilever binder. Subsequent measurements of cadaverine were taken by using the electronic nose when exposed to the same piece of meat for 30 s. Frequency sweeps are performed by the device and read by the appropriate software.

Figure 1 shows cadaverine level readings of two distinct sensors exposed to the same salmon sample for the same amount of time. Surface treatments have not been applied to the cantilever surface prior to binder functionalization, and therefore, might have morphological differences. As it can be seen in the figure, the sensors indicate different cadaverine levels for the same sample.

Figure 2 is optical microscopy images of the functionalized surfaces (cyclam derivative on Si_3_N_4_ surface) without previous surface treatment. Figure 2a,b refer to the pristine binder deposited on the surface without any treatment. Figure 2c,d refer to the binder morphology 24 h after deposition. One can observe binder crystallization and precipitate formation by cyclam clustering. The binder crystallization is a critical feature that should be avoided to ensure reproducible results. Since the cyclam clustering is a random and uncontrollable condition, the cadaverine absorption may vary from sample to sample, which is not desired in industrial applications. Figure 2e,f show the binder morphology after cadaverine exposure. As it can be seen in the picture, the cadaverine modifies the binder properties, and this change is reflected on the cantilever measured signal. Cadaverine absorption varies on the samples due to morphological differences on the binder (edge between binder and surface) but also due to random cyclam crystals formation, and that might be the reason for the distinct readings in Figure 1.

### 3.2. Surface Treatments

Table 1 states the surface energy of the surfaces after the distinct treatments. As described in the method section, the surface energy is obtained by measuring the contact angle of drops of three different reference liquids. All measurements are performed on 5 distinct samples. As it can be seen from the table, the treatments greatly influence the surface energies, but also add a degree of standardization, as reflected by the smaller deviations on surface energy values from the treated samples, in comparison to the reference non-treated samples. The selected physical treatments show an increase in hydrophilicity (increase surface energy), while the explored chemical treatments increase hydrophobicity. The latter might be related to the chemical properties of the chosen chemicals, since both tetraoctylammonium bromide and PBS-tween improve hydrophobicity. A uniform dispersion of nanoparticles on the surfaces is expected to serve as barriers for cyclam diffusion, and the initial hypothesis was that they could either hinder cyclam crystallization or promote it, since crystals could be formed from the nanoparticle-cyclam nucleation sites. The nanoparticle solution included an anti-clustering agent (tetraoctylammonium bromide) to minimize particle clustering.

Both oxygen plasma treatment and annealing, especially at higher temperatures, show to be promising methods for surface properties standardization, since the variation between treated surfaces’ properties is significantly reduced.

Figure 3 illustrates the effect that the different treatments have on surface properties. One can see that the proposed physical treatments increase surface wettability.

Figure 4 illustrates the binder morphologies for the different surfaces, both seen from the top and as seen from the side. As it can be seen from the image, the hydrophobic surfaces have an abrupt border edge, which might vary from sample to sample. On the other hand, the hydrophilic surface has a high wettability, ensuring a smoother interface between the binder edge and the surface, enabling better reproducibility.

From these data, it can be concluded that the suggested physical treatments are the most promising in controlling the morphology, since the binder is deposited uniformly over the surface.

Figure 5 includes images of surfaces treated with nanoparticles (hydrophobic) and plasma asher (hydrophilic) just after functionalization and 72 h after functionalization post-cadaverine exposure.

As it can be seen from the images, the binder deposited on the hydrophobic surfaces has a clear cyclam crystal formation, which appears about 24 h after functionalization, and is still visible after exposure to cadaverine. The same feature was observed on the non-treated samples. However, the crystal formation is not observed on the binder deposited on hydrophilic surfaces, even for longer timeframes. This indicates that cyclam crystallization might come from the binder film internal stress, which is higher in a film from a drop with a high contact angle than film from a drop spread over the surface.

## 4. Conclusions

The objective of this study was to demonstrate a method to achieve controlled surface functionalization of cantilever sensors. Our research hypothesis was that surface energies would play an important role on binder morphology and hinder cyclam crystal formation. We investigated the hypothesis by treating the samples with several modifications. Both suggested physical and chemical treatments have demonstrated to be improving surface properties standardization in relation to non-treated substrates. The proposed chemical methods increase hydrophobicity in a reproducible manner, and the physical methods increase hydrophilicity. In the specific presented case of a cyclam based binder for detection of cadaverine, hydrophilic surfaces ensured reduction of functionalization the drop internal stress and, therefore, crystallization suppression, which is an important aspect for reproducibility of cadaverine absorption. The methods are appliable to cantilevers and assist in the better standardization of cantilever functionalization for industrial applications. The improved technique has resulted in reproducible measurements for the same piece of meat. The devices are now being validated by end users in the meat and fish industry, supermarkets and restaurants.

## Figures and Tables

**Figure 1 sensors-21-06041-f001:**
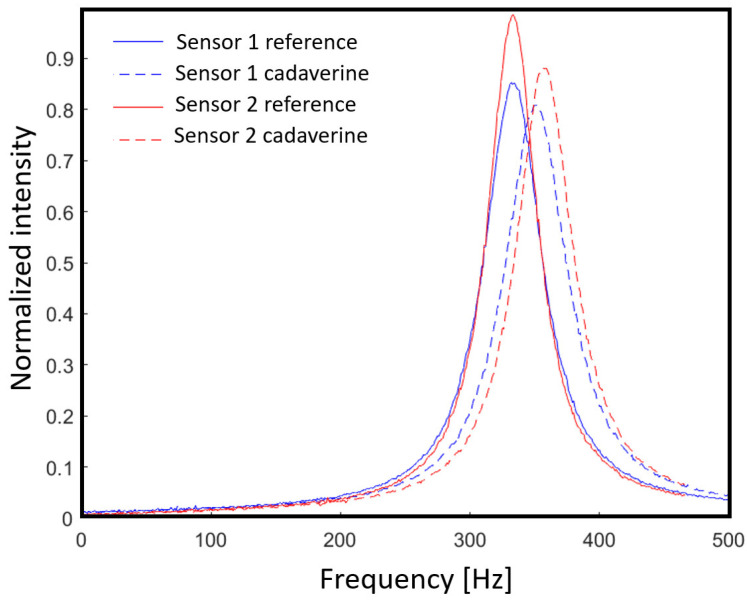
Sensor output shift before and after cadaverine exposition from two distinct sensors with non-standardized binder morphology (clustering).

**Figure 2 sensors-21-06041-f002:**
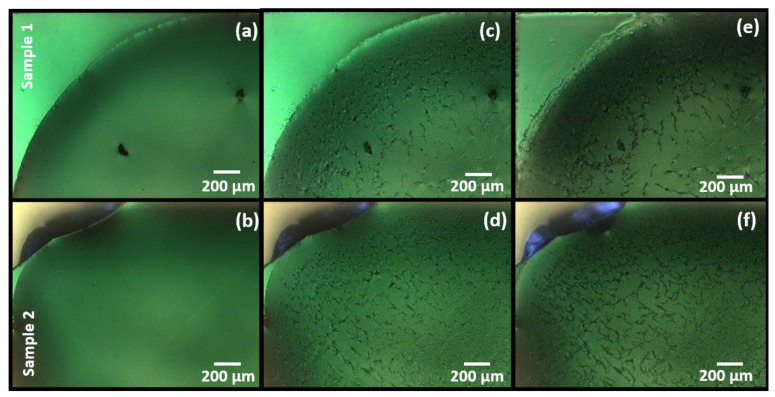
Optical microscopy images of two distinct Si_3_N_4_ surfaces (sample 1 and sample 2) after functionalization (**a**,**b**), after binder crystallization (**c**,**d**) and after exposure to cadaverine (**e**,**f**).

**Figure 3 sensors-21-06041-f003:**
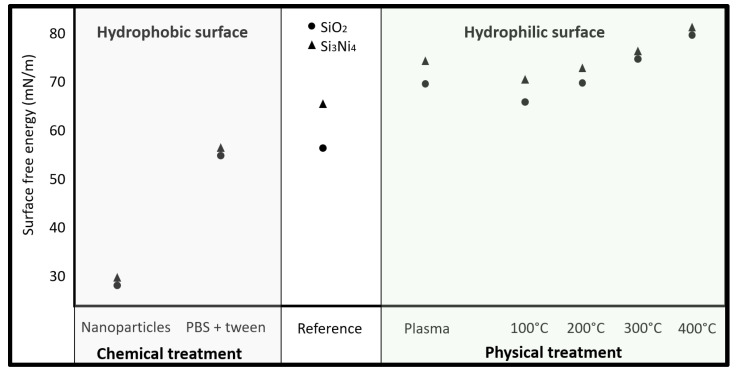
Surface treatments influence on the surface wettability properties.

**Figure 4 sensors-21-06041-f004:**
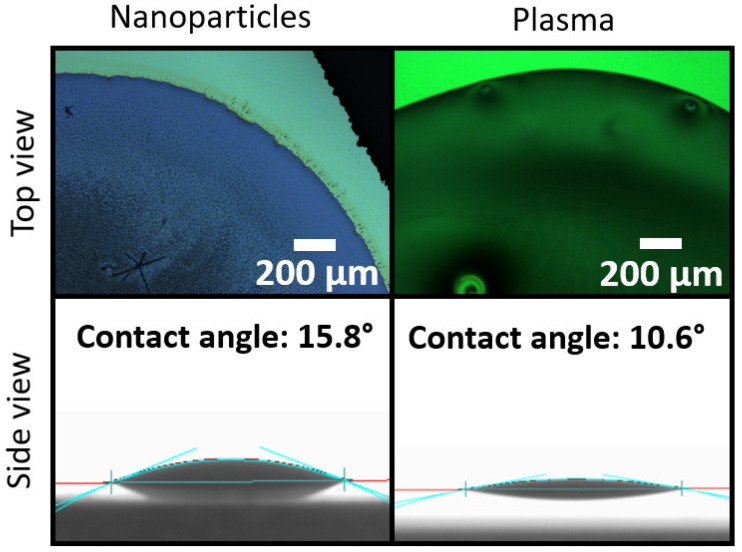
Top-view and side-view images of the binder drop deposited on samples submitted to two different treatments. A clear degree difference of the hydrophilic characteristic is observed.

**Figure 5 sensors-21-06041-f005:**
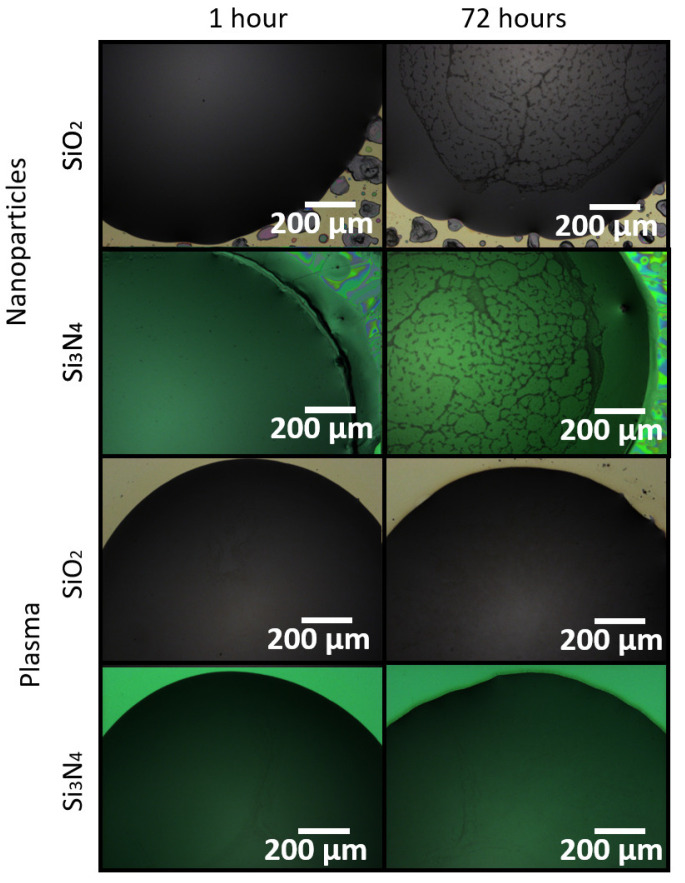
Surfaces treated with the selected treatment just after functionalization (1 h) and 72 h after functionalization, after cadaverine exposure.

**Table 1 sensors-21-06041-t001:** Surface energies after surface treatments.

	SiO_2_	Si_3_N_4_
	Surface Free Energy (mN/m)	Surface Free Energy (mN/m)
No treatment	58.848 ± 2.223	66.258 ± 2.384
Plasma	67.462 ± 0.663	73.948 ± 0.567
PBS tween	57.78 ± 1.223	56.744 ± 1.697
Nanoparticles	29.482 ± 0.854	29.970 ± 0.900
Annealing 100 °C	63.336 ± 1.478	69.158 ± 0.857
Annealing 200 °C	68.420 ± 1.730	71.352 ± 0.826
Annealing 300 °C	73.560 ± 0.993	76.208 ± 1.088
Annealing 400 °C	77.778 ± 0.684	80.734 ± 0.316

## Data Availability

Not applicable.

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
