# Peer review of "Surface Modification Enabling Reproducible Cantilever Functionalization for Industrial Gas Sensors"

_sensors, 2021, doi:10.3390/s21186041_

Round 1

Reviewer 1 Report

The entailed research article on “Surface modification enabling reproducible cantilever functionalization for industrial gas sensors“ by Daniel Mamou et al submitted their research out of cantilever application sensors with detection of a meat freshness volatile biomarker for determination of meat and fish precise expiration dates. Auther’s have described their target on selectivity cantilevers and their need of functionalized with selective binding ability by cyclam-derivative. Base on their biomarkers physical and chemical tunability of surface properties like surface energy directly influences the binder morphology and material clustering and therefore target binding. The article well described their chemical and physical properties and their functionalization influences. Relatively, both physical and chemical treatments have been demonstrated to be improving surface properties standardization concerning non-treated substrates, which more importantly correlated their results outputs.

 The manuscript is well written with supported appropriate citations, characterization/analysis results. Well explained and discussed based on the experimental outputs. I find the manuscript easy going for reading and I recommend the manuscript for publication in your reputed journal “Sensors” and hope this manuscript will support in future for the related research communities.

Author Response

Dear reviewer,

Thanks for the comments. 

Best regards,

Roana

Reviewer 2 Report

In the reviewed work authors performed functionalization of cantilever  by cadaverine selective binder - cyclam-derivative,  to fabricate sensors for detection of a meat freshness.  As chemical and physical surface treatments influence cantilever surface, binder morphology/clustering and overall binding capabilities, binder morphology vs surface energy and cadaverine adsorption was investigated. Since the key issues are the functionalization reproducibility  and target selectivity, authors postulated that in order to avoid or limit  cyclam clustering, the surface should be rendered hydrophilic. In fact, oxygen plasma treatment and annealing were shown to increase hydrophilicity, evidened by an increase in surface energy. The concept of this work is fine, however, some issues need to be further addressed:

- Morphology of the surfaces after functionalization, binder crystallization and exposure to cadaverine should be shown in more detailed way at higher magnification to observe / discuss the effects present at sub-micrometer scale,

- what about the binder crystallization mechanism and crystalline phase formation? is it nucleation-growth mechanism?  Any WAXD data?

- Please discuss the role of nanoparticles.  Are they showing tendency to agglomerate?

- p. 2 – „Plasma treatment: Samples were submitted to oxygen plasma for 30 seconds with a power of 10.2 W.” – how were these conditions selected/optimized?

- What about reproducibility of cadaverine absorption profile? 

- Please add information on sensitivity, resolution, and repeatability of the sensor.

Author Response

Dear Reviewer,

Thanks for your feedback. We are glad to hear that the concept is fine. About the points to be addressed:

  • As reported on our previous publications regarding surface functionalization with cyclam (references 6-8), cyclam crystallization has been a characteristic with several different functionalization parameters and conditions. Reference 6 presents the crystallization mechanisms including high resolution imaging of the cyclam crystals. Cadaverine absorption by cyclam clusters has not been an issue in the proof-of-concept and laboratory tests, since absorption is present and result in a clear change of resonance frequency. However, when we think in upscaling the cantilevers into industrial scale, cyclam morphology comes into play with a crucial role, since crystal structures are forming randomly are difficult to reproduce in a large number of cantilevers. We have added a paragraph on the introduction, where we refer to the previous publications for more information regarding the cyclam clustering mechanisms and high-resolution imaging.
  • The following information has been added to the introduction: Investigations on the influence of functionalization solvents on the crystal formation indicates an important influence on the solvent polarity index on the resulting crystal sizes [6], indicating that high cyclam solubility in solution allows for a disperse material surface distribution upon functionalization, requiring higher diffusion energy for cyclam clustering into large crystals, via a nucleation-growth mechanism. Reference 6 presents the crystallization mechanisms including high resolution imaging of the cyclam crystals.
  • Role of nanoparticles: A uniform dispersion of nanoparticles on the surfaces is expected to serve as barriers for cyclam diffusion, and the initial hypothesis was that they could either hinder cyclam crystallization or promote it, since crystals could be formed from the nanoparticle-cyclam nucleation sites. The nanoparticle solution included an anti-clustering agent (tetraoctylammonium bromide) to minimize particle clustering. This has now been included in the text.
  • Plasma treatment optimization: Samples were submitted to oxygen plasma for 10, 20 and 30 seconds with a power of 10.2 W, which is preset on the available oxygen plasma chamber No significant changes on the surface properties were observed. This has now been included in the text.
  • Reproducibility of cadaverine profile: The improved technique has resulted in reproducible measurements for the same piece of meat. The devices are now being validated by end users on the meat and fish industry, supermarkets, and restaurants. This has now been added to the text.
  • Sensitivity: about 100 Hz/g of cadaverine. Repeatability: The same comment as for the reproducibility. The prototype is now with the end-users.

Best regards,

The authors

Reviewer 3 Report

The article presents a topic about the Surface modification enabling reproducible cantilever functionalization for industrial gas sensors, however there are points that need to be clarified and summarized.

  1. Abstract: I suggest improving the abstract, making clear the purpose and the main results that have been achieved with this study.
  2. Introduction: The introduction is very general and very short. It should be improved and clearly explain the purpose and final objective of this study with appropriate references. In a research paper, it is expected that introduction section briefly explains the starting background and, even more important, the originality (novelty) and relevancy of the study is well established. Once this is done, hypothesis and objectives of the study need to be addressed, as well as a brief justification of the conducted methodology.
  3. Materials and Methods: Explain in greater detail the experimental processes carried out in the research.
  4. Results and discussions: Put in this section only the part of the results obtained in this research for in another separate section (discussion), compare and discuss those results explaining the main advances obtained by comparing and discussing them with the results obtained by the studies of other authors.
  5. Discussion Section: Create a separate Discussion section. The Discussion section should compare the study by clearly comparing the results obtained by the authors with other studies conducted by other authors.
  6. Conclusions Section: Improve the conclusions section, it is very general and does not clearly explain the main objectives achieved in this research. The conclusions section should present in a clear and summarized way the main parts obtained with this study and the main contributions.

Author Response

Dear Reviewer,

Thanks for your feedback. About the points to be addressed:

  • Abstract, we have added: We demonstrated a method for hindering binder functionalization crystallization, leading to reproducible target capture. The results show that cantilever surface treatment is a promising method for achieving a high degree of functionalization reproducibility for industrial cantilever sensors, by controlling binder morphology and uniformity.
  • Introduction: we have added the information as described in your comment.
  • Materials and methods: we have expanded the section.
  • We have attempted to separate the results and discussion section as proposed. However, the result was a more confusing text, where the reader must go back and forth between the sections since we are referring to the same figures in both sections. We would like to keep the original style. Hope that is acceptable.
  • Conclusions: we have expanded the section following your advice.

Best regards,

The authors

Reviewer 4 Report

Dear Editor

 The paper entitled "Surface modification enabling reproducible cantilever functionalization for industrial gas sensors " presents the application of cantilever sensor to  determine meat and fish expiration dates, by functionalization with a cadaverine binder, based on a cyclam-derivative. In particular, the authors investigated the chemical and physical surface treatments influence cantilever performances.

The article is sound and detailed, and I have some comments

1) did the authors observe changes in the consistency of the deposits doo the measurements of the contact angle? That is, does the contact angle value change over time? Are surface deposits (e.g. nanoparticles) partially removed with the measurement?

2) Some acronyms used in the text which should be explained (e.g. AIN, IPA, etc) should be explained

After these small edits the paper in my opinion could be accepted.

Author Response

Dear Reviewer,

Thanks for your feedback. About the points to be addressed:

  • We have measured the contact angle at different time intervals after treatment, and the results did not change over time. However, we have repeated the contact angle measurements in different samples, but never the contact angle again on the same sample, since we believed it could influence the result. Also, the samples which were morphologically investigated are twin samples of the ones where the contact angles were measured.
  • Thanks for the observation, we have clarified the acronyms.

Best regards,

The authors

Reviewer 5 Report

Comments to the authors:

The topic is interesting and well presented. Some corrections should be made.

1) Introduction:

It provides sufficient background and includes all relevant references.

2) Materials and Methods:

Line: 86 (page 2): Explain abbreviations IPA, DI water

Line 93 (page 2): how did you control drying at PBS-tween treatment?

Line 95-97 (page 3): at explanation of surface nanoparticle dispersion: how did you control drying?

3) Results and Discussion:

Results are clearly presented. Few corrections should be made.

Line 171-172 (page 5): correct the bold text of the figure explanation to normal text.

Line 180-181 (page 6): correct the bold text of the figure explanation to normal text.

Line 196-197 (page 7): correct the bold text of the figure explanation to normal text.

Author Response

Dear Reviewer,

Thanks for your feedback. About the points to be addressed:

  • Thanks for that
  • The information has been added into the Materials and Methods section
  • Thanks, the corrections have been done.

Best regards,

The authors

Round 2

Reviewer 2 Report

Authors properly addressed questions and comments raised by the reviewer. The revised version of the manuscript can be published as it is. 

Reviewer 3 Report

Accept in present form